# Ethanolic Extract from Fruits of *Pintoa chilensis*, a Chilean Extremophile Plant. Assessment of Antioxidant Activity and In Vitro Cytotoxicity

**DOI:** 10.3390/plants13101409

**Published:** 2024-05-18

**Authors:** Dioni Arrieche, Andrés F. Olea, Carlos Jara-Gutiérrez, Joan Villena, Javier Pardo-Baeza, Sara García-Davis, Rafael Viteri, Lautaro Taborga, Héctor Carrasco

**Affiliations:** 1Laboratorio de Productos Naturales, Departamento de Química, Universidad Técnica Federico Santa María, Avenida España 1680, Valparaíso 2340000, Chile; dioniarrieche@gmail.com; 2Grupo QBAB, Instituto de Ciencias Aplicadas, Facultad de Ingeniería, Universidad Autónoma de Chile, San Miguel, Santiago 8900000, Chile; andres.olea@uautonoma.cl; 3Centro Interdisciplinario de Investigación Biomédica e Ingeniería para la Salud (MEDING), Escuela de Kinesiología, Facultad de Medicina, Universidad de Valparaíso, Valparaíso 2362905, Chile; carlos.jara@uv.cl (C.J.-G.); juan.villena@uv.cl (J.V.); 4Programa de Conservación de Flora Nativa del Norte de Chile, Biorestauración Consultores, Copiapó 1530000, Chile; javierpardobaeza@japbc.com; 5Instituto Universitario de Bio—Orgánica “Antonio González” (IUBO-AG), Universidad de La Laguna (ULL), 38200 San Cristóbal de La Laguna, Spain; saragarcia50@hotmail.com; 6Escuela de Ciencias Ambientales, Universidad Espíritu Santo, Guayaquil 092301, Ecuador; rviterie@gmail.com

**Keywords:** extremophile plants, *Pintoa chilensis*, cytotoxic activity, antioxidant activity, lipid peroxidation, ROS, *GC–MS* analysis

## Abstract

*Pintoa chilensis* is a shrub with yellow flowers that reach up to two meters high, endemic of the Atacama Region in Chile. This species grows under special environmental conditions such as low altitude, arid areas, and directly sun-exposed habitats. In the present study, ethanolic extract was obtained from fruits of *P. chilensis*, and then partitioned in solvents of increasing polarity to obtain five fractions: hexane (HF), dichloromethane (DF), ethyl acetate (AF), and the residual water fraction (QF). The antioxidant activity of extracts was evaluated by using the DPPH, ABTS, and FRAP methods. The results show that the antioxidant capacity of *P. chilensis* is higher than that reported for other plants growing in similar environments. This effect is attributed to the highest content of flavonoids and total phenols found in *P. chilensis.* On the other hand, the cell viability of a breast cancer cell line (MCF-7) and a non-tumor cell line (MCF-10A) was assessed in the presence of different extract fractions. The results indicate that the hexane fraction (HF) exhibits the highest cytotoxicity on both cell lines (IC_50_ values equal to 35 and 45 µg/mL), whereas the dichloromethane fraction (DF) is the most selective one. The GC–MS analysis of the dichloromethane fraction (DF) shows the presence of fatty acids, sugars, and polyols as major components.

## 1. Introduction

According to the World Cancer Research Fund International, in 2020, about 18.1 million new cases were diagnosed and there were 9.9 million cancer deaths. The most commonly diagnosed cancers are breast, lung, and colorectal, whereas lung, colorectal, and liver are the leading causes of cancer death [1]. These figures indicate that despite the enormous effort displayed to control and prevent cancer, it remains an unsolved health problem worldwide.

Studies over the past six decades have helped to develop different therapy treatments, such as surgery, radiotherapy, immunotherapy, photodynamic therapy, chemotherapy, monoclonal antibodies, and combination chemotherapy.

Chemotherapy is based on small molecules that can act as inhibitors [2,3], antioxidants [4,5,6,7], and apoptosis-inducing agents drugs [2,8,9]. One of the main drawbacks of this therapy is that these anticancer agents target both cancer cells and normal host cells [3]. Nowadays, significant efforts are directed to the development of new bioactive anticancer compounds with greater specificity and enhanced therapeutic efficacy. In this context, the conjugation of therapeutic agents to different ligands has been proven to enhance specific targeted delivery. In addition, the diversity of available ligands allows for the precise tuning of the properties and interactions required for this delivery process [10]. 

Therefore, the search for compounds that act selectively against cancer has increased in recent years. A natural source of bioactive compounds are plants, which have been used by traditional medicine in the treatment of different diseases including cancer. Drugs that have been developed from natural products or the chemical modification of secondary metabolites have been extensively studied and excellent reviews on this matter are available [11,12,13,14,15,16]. The current technologies used in the development of new drugs starting from plants, namely, extraction, isolation, characterization, and modification of lead compounds, have also been reviewed [17,18,19,20,21]. The first step in the process of obtaining natural products from botanical sources is the extraction of plant material by using solvents of varying polarity. This plant extract is then fractionated, and the bioactive compounds are purified and characterized. However, plant extracts have also been used for many years in traditional medicine or ethnopharmacology to treat many infections and diseases. Commonly, crude extracts are a mixture of a huge number of secondary metabolites such as alkaloids, coumarins, phenols, flavonoids, lignans, chalcones, peptides, and simple aromatics that are present in different relative concentrations [22]. Therefore, they could exhibit different biological activities, and in many cases important synergic effects are observed [23,24]. Thus, the obtention of plant extracts to determine the total content of secondary metabolites and the screening of their biological activities have been subjects of increasing interest [16,23,25,26,27,28,29,30,31,32,33,34,35].

On the other hand, it is well established that secondary metabolites are involved in plant protection mechanisms from abiotic and biotic stresses [36,37]. Therefore, plants that inhabit desertic environments have developed mitigation systems to adapt themselves to live under extreme conditions, such as drastic temperature changes between day and night, strong ultraviolet radiation, and water scarcity. One of these defense mechanisms is the production of different chemical products, with interesting bioactivities, namely antioxidant, antimutagenic, antibacterial, and UV-protective [38,39,40,41]. The Atacama Desert, located in the north of Chile and framed by the Andes Mountains, it is considered the driest desert on Earth and one of the most extreme environments on the planet. However, it is estimated that more than 2500 species of plants grow in this environment, and more than half are endemic. Among them, there are some plants belonging to the *Zygophyllaceae* family, which comprises 22 genera and 285 species and are widespread globally [42]. Extensive work has been performed on the bioactivity of their extracts as a function of the phytochemical content, which for the same species varies with its geographical location [43,44,45,46,47,48]. Most of these studies have been carried out with *Zygophyllaceae* plants from Asia, Africa, and Europe [49]. This knowledge of plants in South America is far more restricted [50,51,52]. In Chile, five species of this family are known: *Bulnesia chilensis*, *Fagonia chilensis*, *Larrea nitida*, *Porlieria chilensis*, and *Pintoa chilensis*. From these, *P. chilensis* is a shrub with yellow flowers up to two meters high, endemic in the Atacama region. Although there is no information about the pharmacological use of this plant, there are ethnocultural antecedents transmitted by the indigenous tribes of northern Chile. These come mainly from the Aymara people and indicate that an aqueous infusion obtained from fruits and leaves has been used preferentially for the treatment of abdominal and bone pain. The scarce antecedents available, as well as the fact that this shrub survives in extreme environments, suggest that this plant could be a source of new metabolites with interesting biological activities.

Herein, we described the antioxidant activity and cytotoxicity in vitro of ethanolic extracts of the fruits and aerial parts of *P. chilensis*, as well as the identification of the fraction with the most potential for the discovery of its bioactive compounds.

## 2. Results and Discussion

### 2.1. Extraction Procedure

Extracts were obtained by the maceration of dried vegetal material with an ethanol/water mixture followed by ultrasound-assisted extraction. This extraction procedure led to yields of ethanolic extracts of 43.8% for aerial parts (EF1) and 30.45% for fruits (EF2) (Appendix A and Appendix A). 

### 2.2. Total Phenolic (TPC) and Total Flavonoid (TFC) Content

The antioxidant role of phenolic compounds has been established, and therefore its presence in plant extracts has often been linked to a wide array of therapeutic and physiological benefits [53]. On the other hand, flavonoids have shown in both in vitro and in vivo studies to possess anti-inflammatory, immunomodulatory, and anticancer properties [54,55,56]. Thus, to assess the total content of phenols and flavonoids in the ethanolic extracts of *P. chilensis*, colorimetric methods were employed, as detailed in the Section 3. Briefly, the concentrations of phenols and flavonoids were determined through spectrophotometric measurements, utilizing absorbance calibration curves derived from gallic acid and quercetin, respectively. Consequently, the results are expressed as equivalents of these standards (mg/g of dried plant material) (Table 1).

The total phenol and flavonoid contents showed significant differences between extracts from the aerial parts and fruits, i.e., the EF1 total phenolic content was higher than that found for EF2, whereas the reverse situation was found for the total flavonoid content (*p* < 0.05). Even though this is the first report on *P. chilensis*, there have been some previous studies on other plants from the family of *Zygophyllaceae* spp. growing in similar conditions in the north of Chile and northwestern Argentina. Thus, phenolic compounds and lignans have been reported in *Porlieria chilensis* [52], *Larrea nitida* [57,58], and *Larrea divaricate* [50]. In the latter, the measurements of phenols and flavonoids were performed relative to caffeic acid and rutin, respectively. Thus, these results cannot be compared directly with those reported herein. However, the ratio of phenols/flavonoids found for EF1 (5.21) and EF2 (2.04) was different from that reported by Dadé et al. (2.82) [50]. In other desertic regions, plants of *Zygophyllaceae* spp. have been extensively studied [49] and the results for species from the fagonia genera [46,47] and tribulus [43,48] have been reviewed. More specifically, extracts of *Fagonia arabica*, *F. mollis*, and *F. creticus* plants collected in Egypt, whose geographic conditions are like those of the Atacama Desert, exhibited TPC values between 4.38–9.62 mg GAE/g DW, whereas TFC values between 5.11–7.16 mg QE/g DW were obtained [59]. Similarly, a study of *Zygophyllum album* aerial and root extracts yielded TPC values equal to 3.86 and 5.26 mg GAE/g DW, and TFC values of 2.33 and 1.36 mg QE/g DW, respectively [60].

On the other hand, TPC and TFC have also been reported for plants from different families that inhabit the altiplano of northern Chile. For example, the study of *Prosopis chilensis* (Algarrobo) has found TPC values between 8.2–25.7 mg GAE/g DW, well below those reported in this work for *P. chilensis* (EF1 and EF2), and TFC values between 1.7–5.6 mg QE/g DW [61].

Thus, these results indicate that ethanolic extracts of *P. chilensis* contain high amounts of both phenolic and flavonoid compounds. This is quite an interesting result because it has been demonstrated that antioxidant and biological activities are correlated with TPC and TFC [62,63]. 

### 2.3. Antioxidant Activity

Antioxidant activity is the result of trapping radicals involved in the oxidation process of a substrate. Thus, the antioxidant will exert a protective effect on a substrate only when this reaction is faster than that between the radicals and substrate. Natural extracts are a complex mixture compounds that can react with peroxyl radicals, i.e., phenols and flavonoids. Therefore, the antioxidant activity of plant materials can be attributed to the competitive reactions of various molecules with these radicals. Thus, total antioxidant capacity (TAC) has been defined and used to measure the antioxidant effects of extracts from plant material. Commonly, TAC is expressed as equivalents of Trolox, and this parameter is termed total equivalent antioxidant capacity (TEAC) [64]. To assess the antioxidant capacity of molecules and natural antioxidants, some direct and indirect methods have been described [65,66]. The latter are the most common and have been used to determine antioxidant properties in plant extracts [31,67,68,69]. Thus, the antioxidant activity of EF1 and EF2 extracts was evaluated using 1,1-diphenyl-2-picryl-hydrazyl (DPPH), 2,2′-azino-bis-3-ethylbenzothiazolin-6-sulphonic acid (ABTS), and ferric-reducing antioxidant power (FRAP) assays. The results obtained for both extracts are listed in Table 2 and are expressed in different units depending on the method used: IC_50_ (mg/mL) for DPPH; mM of Trolox equivalent antioxidant capacity (TEAC) for ABTS and FRAP.

It is interesting to compare the antioxidant activity of *P. chilensis* with that reported for Zygophyllaceae family plants inhabiting sites with similar environmental conditions. For instance, applying the DPPH method to extracts derived from *Fagonia arabica*, *F. mollis*, and *F. criticus*, plants from the Egyptian Desert, revealed IC_50_ values ranging between 740 to 820 mg/L [59]. In contrast, an IC_50_ of 291.9 mg/L was reported for a methanol extract from the aerial parts of a *Zygophyllum album* plant, also sourced from Egypt [60]. Similarly, methanolic and water/methanol extracts from *Tribulus terrestris*, harvested in southern Pakistan, displayed IC_50_ values of 71.4 mg/L and 141.2 mg/L, respectively [44]. These values indicate that the antioxidant capacity of these species is much lower than that reported herein for *P. chilensis*.

Finally, the same trend is observed by comparing the antioxidant activity of *P. chilensis* with other Chilean plants living in desertic areas. Namely, the highest IC_50_ values, ranging from 23.74–100 mg/L, as compared with the EF1 extract, were reported for six samples of *Proposis chilensis* from northern Chile [61].

### 2.4. Cytotoxic Activity

The cytotoxic activity of the EF1 and EF2 extracts of *P. chilensis* was evaluated in vitro against two cancer cell lines, namely MCF-7 breast cancer and HT-29 colon cancer. A non-tumoral cell line, human breast epithelial MCF-10A, was used as a control. IC_50_ values were determined for each cell line by measuring cell viability in the presence of different extract concentrations (0–200 µg/mL). The results, summarized in Table 3, indicate that the EF2 extract exhibited a moderate cytotoxic activity against breast cell line MCF-7, and had no significant cytotoxic effect on the other cell lines. On the other hand, the high IC_50_ values obtained for EF1 (>200 µg/mL) indicate that this extract can be considered inactive for all cell lines. 

It has been shown that reactive oxygen species (ROS) play an essential role in the signal transfer, cell proliferation, and revascularization of tumor cells [5]. A gradual ROS increase can also promote the cell proliferation, diffusion, and metastasis of tumors. On the other hand, a high level of ROS will damage the DNA of cancer cells, resulting in cell death [5]. Thus, ROS levels produced by the treatment of tumoral cells with the EF2 extract were quantified by flow cytometry using the DCFH_2_-DA ROS detection assay [70]. The results shown in Figure 1 (Appendix A). indicate that in the absence of any treatment, ROS production by normal cells was higher than that found for the MCF-7 cell line. This difference was even larger in the presence of daunorubicin (DNR), which was used as a positive control. Interestingly, this relation was reversed in cells treated with the EF2 extract. On one side, for non-tumor cells, MCF-10A, a ten-fold decrease in ROS production was observed, suggesting an antioxidant effect of the EF2 extract. On the other hand, ROS production by MCF-7 cells strongly depended on the EF2 concentration, i.e., at 50 and 100 µg/mL it increased to levels much higher than those observed for DNR, whereas at 200 µg/mL a sharp decrease was observed. These results indicate that the EF2 extract affected the ROS production of MCF-7 cells, acting either as a pro-oxidant at low concentrations or an antioxidant at higher concentrations. Even more interesting is the comparison of ROS production in MCF-10 and MCF-7 in the presence of low concentrations of the EF2 extract. For normal cells, EF2 exhibited a strong antioxidant effect, whereas for cancer cells, EF2 acted as a pro-oxidant, suggesting that EF2 could promote apoptosis in MCF-7 cells. This dual effect has been previously observed for eugenol and other organic molecules [71,72]. The antioxidant effect is associated with the inhibition of cancer development in non-tumoral cells, while the pro-oxidant effect has been related to the death of cancer cells through various signaling pathways [73].

On the other hand, it has been demonstrated that lipid peroxidation plays an important role in human diseases, such as inflammation, cancer, and neurodegenerative processes [74,75,76,77]. This process, mediated by ROS, generates deleterious peroxide products, i.e., endoperoxides or hydroperoxides, which induce toxic effects through a mechanism that involves the alteration of the cellular membrane or, alternatively, producing reactive species capable of interconnecting DNA and proteins [78]. Additionally, lipid peroxidation is very important to cell viability, because changes in the assembly, structure, and dynamics of cell membranes induce cell death, probably via apoptosis [78,79,80].

Therefore, to evaluate oxidative damage on the cell membranes of MCF-7 and MCF-10A cell lines, caused by the EF2 extract, a fluorescence assay was used. This assay measures changes in the fluorescence intensity of BODIPY-C11, a lipophilic fluorescence sensor that reacts rapidly with ROS [81]. Thus, the decrease in fluorescence can be correlated to lipid peroxidation occurring in cell membranes. In Figure 2 (Appendix A), the results are presented as the percentage of stained cells that suffered lipid peroxidation in the presence and absence of different concentrations of the EF2 extract. 

This figure shows that the lipid peroxidation of the MCF-7 cell line, compared to the positive control DNR, changed with DF concentration, i.e., increased at 50 and 100 µg/mL and decreases at 200 µg/mL. On the other hand, for the non-tumor cell line MCF-10A, a decrease in lipid peroxidation was observed for all the tested concentrations of the EF2 extract. These results are consistent with the ROS production measurements shown in Figure 1. In non-tumoral cells, EF2 seemed to act as an antioxidant agent, thereby protecting the cell against peroxidative damage. Probably, this effect was due to the activation of cell antioxidant mechanisms of defense, both enzymatic and non-enzymatic [82]. These defense mechanisms are generated when ROS levels exceed the cell’s tolerance levels, which generates the transcription of genes linked to the expression of antioxidant enzymes, thereby causing a decrease in ROS levels at the cellular level [83,84]. For the MCF-7 cancer cell line, our results suggest that EF2 induced lipid peroxidation, probably promoting an apoptotic mechanism, which was possibly caused by a change in the cell membrane. 

Mitochondria play an important role in energy metabolism and are also essential in the process of programmed cell death [85,86,87]. More specifically, changes in the mitochondrial membrane potential are a distinctive marker of early apoptosis [88]. Therefore, it is possible to assess apoptosis by determining changes in the mitochondrial membrane potential (ΔΨmt). Thus, changes in the mitochondrial membrane potential were assessed by using flow cytometry with rhodamine-123 (Rh-123) [89]. The results, shown in Figure 3 (Appendix A), indicate a reduction in the mitochondrial membrane potential of the MCF-7 cell line treated with EF2 for all tested concentrations. This effect is like that observed with daunorubicin, which is known to induce apoptotic processes, independent of EF2 concentration.

Interestingly, for the non-tumoral cell line, MCF-10A, it was observed that the mitochondrial membrane potential remained intact Figure 3 (Appendix A), so its permeability was not affected. This result suggests that EF2 exhibited a selective apoptotic effect on the cancer cell line, with no effect on normal cells. 

Apoptosis is one of the most important and most studied pathways of cell death. This process is characterized by a series of morphological changes that are regulated by caspases [90,91]. The activation of these proteins generates the rupture of critical cellular substrates, which generate profound morphological changes at the cellular level.

Thus, to verify that the EF2 extract was acting through an apoptotic mechanism on the cancer cell line, active caspases were detected by flow cytometry by using a fluorescent FITC-VAD-FMK inhibitor of caspases, which binds irreversibly to activated caspases inside the cell. The results, shown in Figure 4, are given as the percentage of cells stained with this apoptosis marker. The data in Figure 4 show that there was a decrease in caspase activation for the MCF-7 cell line, at all tested concentrations, compared to the positive control (Appendix A). This result indicates that apoptosis was not involved in the cytotoxicity effect induced by the EF2 extract. Currently, around thirteen mechanisms of cell death are known [92], and EF2 probably acted through one of these pathways. However, for the non-tumoral cells, an increase in caspase activation was observed at the highest tested concentration of EF2 (Appendix A). Probably, at high concentrations, EF2 promoted the activation of inflammatory caspases involved in other metabolic process at the cellular level.

Finally, considering that chemotherapy drugs target both cancer cells and normal host cells, selectivity towards a specific cancer cell line is a fundamental feature to reduce undesirable side effects. The selectivity index is essential to identify a substance with promising biological activity and negligible cytotoxicity [93]. Consequently, extract EF2 (SI > 1.8 for MCF-7) appears to be a potential candidate to identify cytotoxic metabolites for specific cancer cell lines.

### 2.5. Cytotoxic Activity of Fractions to EF2

Extract EF2 was resuspended in a mixture of water–ethanol (9:1) and fractionated with hexane (HF), dichloromethane (DF), and ethyl acetate (AF); four organic fractions and a residual aqueous fraction (QF) were obtained (Appendix A). The effect of each fraction on the viability of MCF-7 and MCF-10A was evaluated, using the sulforhodamine B colorimetric assay.

The results, shown in Table 4, indicate that fractions HF, DF, and AF decreased the cell viability of both cell lines, with the IC_50_ ranging between 35.3–232.5 µg/mL. The highest effect was shown by HF (IC_50_ = 35.3 for MCF-7), while DF had a lower activity but presented the highest selectivity. For this reason, we chose this fraction for further chemical study.

### 2.6. GC–MS Analysis

The dichloromethane fraction was analyzed by GC–MS and the results are shown in Table 5 (Appendix A and Appendix A). 

Three groups of compounds were identified in this fraction, namely, fatty acids: palmitic acid, stearic acid, myristic acid, and 2-monostearin, totaling 3.37%; sugars: sucrose and β-D-allopyranose, totaling 2.21%; and polyols: 1,3-propanediol, D-pinitol, and myoinositol, totaling 8.76% of the total DF composition. 

Palmitic and stearic acids have been identified in other plants belonging to the *Zygophyllaceae* family: *Zygophyllum fabago*, *Z. album*, *Z. luntii*, *Tribulus terrestris*, *Peganum harmala*, and *Nitraria sibirica* [94,95,96,97,98]. It has also been shown that these acids exhibit inhibition effects on the cell proliferation of various cancer cell lines [99,100]. More specifically, the cytotoxicity effects of palmitic acid are due to a non-apoptotic mechanism pathway [99,101,102], which is in line with the experimental results obtained for the *P. chilensis* extract. On the other hand, the effect of stearic acid is not entirely clear, but there are some studies suggesting that this fatty acid probably alters the rigidity of tumor cell membranes, thereby causing cell death. This effect has been reported for other natural extracts in which this acid has also been identified [103,104]. Thus, these results suggest that the anticancer activity of the DF fraction could be associated with the presence of palmitic and stearic acids.

In the same context, it has been shown that sugars, such as β-D-allopyranose, and polyols, such as D-pinitol or myoinositol, exhibit effects on cell growth by inducing cell death [105,106,107]. Therefore, these compounds might also contribute to the observed activity of DF. Probably, in the case of β-D-allopyranose, its effect could be the consequence of a different mechanism, such as oxidative stress, arrested cell proliferation, enhanced autophagy, apoptosis, or angiogenesis inhibition [108]. Similar behavior has been observed for other sugars such as glucopyranose or sugar polymers such as pectin [109]. D-pinitol exerts its anti-cancer activity by inducing the blocking of the NF-ⱪB factor pathway [110,111]. Regarding myoinositol, there are several studies that show that this compound is capable of suppressing tumor growth both in vitro and in vivo [107,112,113]. These studies seem to indicate that the DF cytotoxic effect could be due to the presence of these compounds, causing cell death through a non-apoptotic pathway.

Therefore, the cytotoxic activity exhibited by the DF extract could be attributed to the presence of palmitic acid, stearic acid, D-pinitol, and myoinositol acting by themselves or in a synergistic way. The synergic effect of natural products has been demonstrated in several studies [24,114]. Nevertheless, to confirm a synergic effect, it is necessary to carry out additional studies using pure compounds and their mixtures to confirm that the activity of the latter is enhanced compared to the activity of pure compounds.

## 3. Materials and Methods

### 3.1. Plant Collection and Identification

Aerial plants and fruits of *P. chilensis* Gay were collected from Copiapó, Atacama province, Chile (27°31′35.55″ S. 70°15′57.35″ O) in November 2020. The species was recognized by the forestry engineer Javier Pardo Baeza of the biorestoration consultants and identified by the botanist Gloria Rojas Villegas. A voucher specimen (No. SGO 170532) was deposited at the Natural Museum of Natural History of Santiago, Santiago, Chile (Appendix A).

### 3.2. Extraction Procedure and Liquid–Liquid Fractionation

The dried and powdered aerial tissues (EF1) and fruits (EF2) of *P. chilensis* Gay (50 g) were macerated with ethanol–water (7:3) under constant agitation using an orbital shaker (150 rpm) at 25 °C for 96 h. Then, ethanolic extracts from the obtained vegetal material were exhaustively extracted by sonication three times over 1 h at 50 °C. All solvents used were of chromatographic grade. The resulting mixture was filtered through Whatman No. 1 filter paper (Sigma-Aldrich, Darmstadt, Germany) and dried under reduced pressure with a rotary evaporator (Rotavapor R-300, Büchi, Barcelona, Spain). To eliminate water, it was freeze-dried, obtaining the ethanolic extracts EF1 and EF2 (black viscous solid, 21.9 g, and gray solid, 15.3 g, respectively). Then, 15 g and 12 g of ethanolic extracts EF1 and EF2 were suspended in 300 mL of a mixture of ethanol–water (1:9) and partitioned with n-hexane, dichloromethane, ethyl acetate, and a residual aqueous fraction. Extracts and fractions were stored at −20 °C until required for analysis (Appendix A and Appendix A).

### 3.3. Phytochemical Analysis of Extracts

#### 3.3.1. Determination of Total Phenol Content

The total phenol content of ethanolic extracts was determined using the Folin–Ciocalteu reagent [105]. Gallic acid was used as a reference and a calibration curve was made (20–100 µg/mL). A volume of 500 µL of ethanolic extracts (EF1 and EF2) were combined with 2.5 mL of Folin–Ciocalteu reagent, which had been diluted 1:10 with distilled water, and then neutralized with 2 mL of sodium carbonate solution (7.5%, *w*/*v*). This reaction mixture was then incubated in darkness for 2 h at room temperature. Subsequently, the absorbance was measured at 700 nm using a UV spectrophotometer (UV Analyst-CT 8200). The total content of phenolic compounds was determined using a linear regression equation derived from the calibration curve of gallic acid. The content of total phenolic compounds was calculated as mean ± SD and expressed as mg gallic acid equivalent (GAE)/g dry weight based on the calibration curve [105]. 

#### 3.3.2. Estimation of the Total Flavonoid Content

Using the method described by Madaan, the content of total flavonoids in ethanol extracts (EF1 and EF2) was determined [106]. Quercetin was used as to make a standard calibration curve. In a test tube, 1 mL of 2% aluminum chloride and 1 mL of ethanolic extract were mixed. After incubation for 15 min at room temperature, the absorbance at 430 nm was determined by using a UV–Vis spectrophotometer (UV Analyst-CT 8200). A mixture of all reagents except aluminum chloride, which was substituted with an equivalent volume of distilled water, was used as a reference. The quantity of flavonoids was determined using a linear regression equation obtained from the quercetin calibration curve. Experiments were performed in triplicate and expressed as mg quercetin (QE)/g dry weight [106].

### 3.4. Measurement of Antioxidant Capacity 

#### 3.4.1. DPPH-Radical-Scavenging Assay

Free-radical-scavenging capacity of extracts was assessed using a method initially proposed by Brand-Williams and later modified by Miliauskas [27,65]. Briefly, ethanolic extracts were dissolved in ethanol (ranging from 0 to 10 mg/mL). Subsequently, 100 µL of each solution was mixed with DPPH solution (2.9 mL, 50 µM). A control solution containing 100 μL of ethanol was also prepared. Both samples and control solutions were then incubated for 15 min at room temperature, after which the absorbance of DPPH radicals was measured at 517 nm. The radical-scavenging capacity was determined using the following equation:$$RSC=\%DPPH\;radical\;scavenging=\;\left[\left(A_{C}-A_{S}\right)/A_{C}\right]\times 100$$
where *A_C_* and *A_S_* are the absorbances of the control and extracts, respectively. Graphing *RSC* values against extract concentration and fitting the data to a dose–response equation provides IC_50_ values listed in Table 2. IC_50_ values represent the extract concentration needed to scavenge 50% of the DPPH radicals.

#### 3.4.2. ABTS

The total radical-trapping antioxidant potential (TRAP) was evaluated by ABTS assay [115,116]. Equal volumes of a 10 mM ABAP (2,2′-azo-bis (2-amidino propane) solution and a 150 μM ABTS (2,2′-azinobi (3-ethylbenzothiazoline-6-sulphonic acid) solution were mixed using PBS 100 mM at a pH of 7.4 (TRAP solution), and then incubated at 45 °C for 30 min. ABAP induces the formation of relatively stable ABTS radical cations, which can be determined spectrophotometrically at 734 nm. Addition of ethanolic extracts (10 μL, 1.0 mg/mL) to the TRAP solution (990 μL) results in trapping of ABTS^∙+^, and the decrease in absorbance was measured after 30 s at room temperature using ABTS solution as a reference. The total antioxidant capacity (TRAP) of ethanolic extracts was quantified as mM Trolox equivalents (TEAC), utilizing a standard curve ranging from 0 to 120 mg/mL Trolox. Each measurement was repeated three times.

#### 3.4.3. Ferric-Reducing Antioxidant Potential Assay (FRAP)

The ability of extracts to deactivate radicals by electron transfer was assessed by FRAP assay, which is a method used to determine the ferric-reducing potential of antioxidant compounds [31]. In this method, the reduction of a ferric to a ferrous complex by the action of electron-donating compounds is determined by measuring the absorbance at 593 nm. The FRAP reagent is prepared by mixing 10 volume of 300 mM acetate buffer, pH 3.6, with 1 volume of 10 mM TPTZ (2,4,6-tri(2-pyridyl)-s-triazine) in 40 mM hydrochloride acid and 1 volume of 20 mM ferric chloride. Extracts (100 µL, 1 mg/mL) were combined with 300 µL of deionized water and 3 mL of FRAP reagent. The resulting mixture underwent incubation for 30 min at 37 °C in a water bath. After this period, the absorbance was measured at 593 nm, with ethanol serving as a reference. FRAP values were determined by calculating the absorbance difference between sample and reference. FRAP values were reported in mM Trolox equivalents. All measurements were conducted in triplicate. 

### 3.5. Cell Viability

#### 3.5.1. Cell Culture

Human cancer cell lines HT-29 (colon), MCF-7 (breast), and a non-tumoral human cell line MCF-10A (mammary epithelial cell) were maintained in Ham’s-F12: DMEM high-glucose medium (Gibco, San Diego, CA, USA), supplemented with 10% (*v*/*v*) fetal bovine serum (FBS) at 37 °C in a humidified 5% CO_2_ incubator. Appropriate volumes of the working solutions were added to the medium to reach the indicated concentrations (0, 25, 50, 100, and 200 µg/mL), and cells were then incubated for the indicated periods of time.

#### 3.5.2. In Vitro Growth Inhibition Assay

The sulforhodamine B assay was used according to the method of Vichai and Kirtikara [117]. Extracts were dissolved just before use in 1% ethanol. Briefly, 3 × 10^3^ cells/well were set up in a 96-well and flat-bottomed microplate (200 µL). Cells were incubated at 37 °C in a humidified 5% CO_2_/95% air mixture and treated with extracts at different concentrations (0, 25, 50, 100, and 200 µg/mL) for 72 h. The cells that received only the medium containing 1% ethanol were used as a control. Following drug exposure, cells were fixed with 50% trichloroacetic acid at 4 °C. After rinsing with distilled water, cells were stained with 0.1% solution of sulforhodamine B (Sigma-Aldrich, St. Louis, MO, USA) in 1% acetic acid (50 µL/well) for 30 min. Subsequently, cells were washed with 1% acetic acid to remove unbound stain. The protein-bound stain was solubilized with 100 µL of 10 mM unbuffered tris-base, and cell density was determined using a microplate reader (wavelength 540 nm). The presented values represent the mean ± SD of three independent experiments performed in triplicate. The half-maximal inhibitory concentration (IC50) was calculated using Sigmaplot 11.0 [118].

#### 3.5.3. Determination of Mitochondrial Membrane Permeability by Flow Cytometry

Rhodamine-123, a cationic voltage-sensitive probe known for its reversible accumulation in mitochondria, was employed to monitor alterations in mitochondrial membrane potential. Cells were treated with ethanolic extracts (50, 100, and 200 mg/mL) for 48 h, followed by staining with rhodamine-123 (1 µM). Stained cells were then incubated in darkness for 1 h at 37 °C. Subsequently, the medium was aspirated, and cells were washed twice with PBS. Cells were then trypsinized and collected by centrifugation (10 min at 1500 rpm). The supernatant was discarded, and the cell pellet was resuspended in PBS and analyzed by flow cytometry using the filter FL1. Results are expressed as the percentage of cells stained with rhodamine-123 [32,119] (Appendix A).

#### 3.5.4. Determination of Lipid Peroxidation by Flow Cytometry

BODIPY-C11 is a fluorescent probe used to determine lipid peroxidation and antioxidant efficacy in model membrane systems and living cells. Cells were incubated with ethanolic extracts (50, 100, and 200 mg/mL) for 48 h, and subsequently stained with BODIPY-C11 (2 µM) and incubated in darkness for 30 min at 37 °C. Then, medium was removed, and cells were washed twice with PBS. Later, cells were trypsinized and collected by centrifugation (10 min at 1500 rpm). The supernatant was discarded, and the cell pellet was resuspended in PBS and analyzed by flow cytometry using the filter FL1. Results are expressed as the percentage of cells stained with BODIPY-C11 (Appendix A).

#### 3.5.5. Determination of ROS by Flow Cytometry

DCFH_2_-DA (2,7-dichloro-dihydro-fluorescein acetate) is the most common and sensitive probe in detecting intracellular ROS. Cells were incubated with ethanolic extracts (50, 100, and 200 mg/mL) for 48 h, and subsequently stained with DCFH_2_-DA (1 mM) and incubated in darkness for 1 h at 37 °C. Then, medium was removed, and cells were washed twice with PBS. Later, cells were trypsinized and collected by centrifugation (10 min at 1500 rpm). The supernatant was discarded, and the cell pellet was resuspended in PBS and analyzed by flow cytometry using the filter FL1. Results are expressed as the percentage of cells stained with DCFH_2_-DA [118] (Appendix A).

#### 3.5.6. Determination of Caspase Activation by Flow Cytometry

Caspase activity was evaluated by using a fluorescent inhibitor of caspases labeled with fluorescein isothiocyanate (FITC-VAD-FMK). The CaspACE FITC-VAD-FMK In Situ Marker was procured from Promega. In brief, cells were treated with ethanolic extracts (50, 100, and 200 mg/mL) for 48 h. Then, cells were incubated with CaspACE FITC-VAD-FMK (5 µM) in darkness for 30 min at room temperature. After incubation, the medium was aspirated, and cells were washed twice with PBS. Treated cells were harvested by trypsinization and centrifugation (10 min at 1500 rpm). The supernatant was discarded, and the cell pellet was resuspended in PBS and analyzed using flow cytometry with the FL1 filter. Results are presented as the percentage of stained cells [32,120] (Appendix A).

### 3.6. Gas Chromatography–Mass Spectrometry (GC–MS) Analysis

Dichloromethane fraction (DF) of *P. chilensis* was analyzed using gas chromatography–mass spectrometry equipment from Agilent Technologies (Santa Clara, CA, USA, 7890A GC system and 5975C inert XL MSD with triple-axis detector). DF (1 mg) was mixed with 100 µL BSTFA in a water bath to 80 °C for 2 h. Then, 2.0 μL of sample was injected at 250 °C in splitless mode into a DB-5MS column (30m × 0.25 mm) with helium as the carrier gas (1.2 mL min^−1^). The detector temperature was set at 280 °C, while the oven temperature started at 70 °C for 2.0 min and then ramped up to 285 °C at a rate of 5 °C per min. Electron ionization at 70 eV and 230 °C was utilized as the ion source, and data compounds were acquired using full scan mode (40–1000 amu). Identification of compounds was performed by comparing their retention index and mass spectra with data from the Wiley 9th with the NIST 2011 MS Library database. Additionally, retention indices were determined using a series of n-alkanes (C_7_–C_40_) (Appendix A and Appendix A).

### 3.7. Statistical Analysis

All data were expressed as the mean ± standard deviation (SD). Due to non-parametric data, Kruskal–Wallis ANOVA was used with the STATISTIC 7.0 program. Statistical significance was defined as *p* < 0.05.

## 4. Conclusions

This study presents, for the first time, the in vitro biological activity of ethanolic extracts and fractions obtained from the extremophile and monotypic plant of northern Chile *Pintoa chilensis*. The fruit extract (EF2) contained a higher content of flavonoids, whereas the total phenol content showed no significant differences between both extracts EF1 and EF2. The results obtained showed that the EF2 extract was selective against the MCF-7 cell line compared to the non-tumorous MCF-10A cell lines. Additionally, one can infer that the extract caused cell death in the tumor cell line by a non-apoptotic mechanism because caspase activation was not observed. As a result of the findings obtained, it is necessary to further investigate the mechanisms of regulated cell death that are being executed because of EF2 exposure, as well as to perform future studies to identify the types of caspases that are being produced in the non-tumor cell line.

## Figures and Tables

**Figure 1 plants-13-01409-f001:**
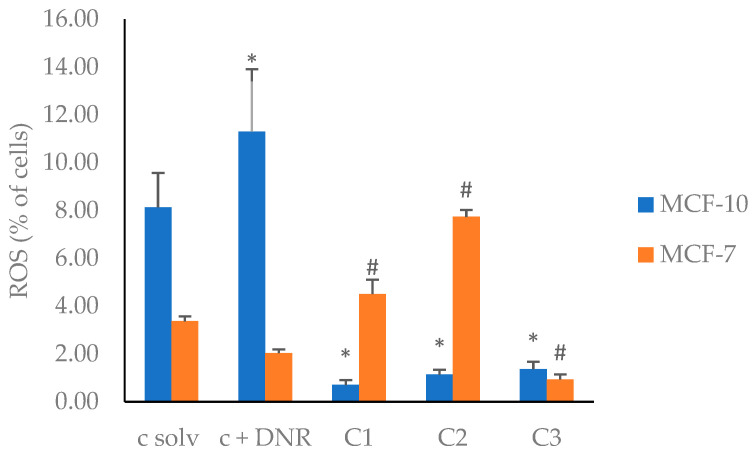
ROS production in MCF-7 and MCF-10A cell lines induced by EF2 extract. c solv: solvent control (EtOH); c + DNR: daunorubicin; C1: 50 µg/mL; C2: 100 µg/mL; C3: 200 µg/mL. *, #: different symbols correspond to significant differences among treatments and c solv (*p* < 0.05).

**Figure 2 plants-13-01409-f002:**
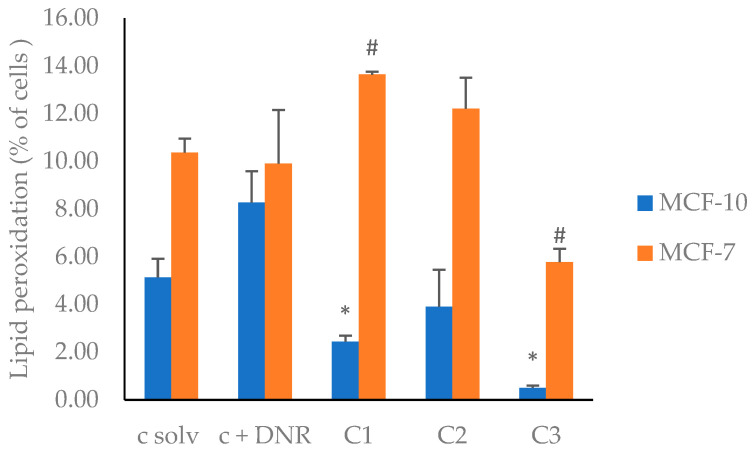
Lipid peroxidation in MCF-7 and MCF-10A cell lines induced by EF2 extract. c solv: solvent control (EtOH); c + DNR: daunorubicin; C1: 50 µg/mL; C2: 100 µg/mL; C3: 200 µg/mL. *, #: different symbols correspond to significant differences among treatments and c solv (*p* < 0.05).

**Figure 3 plants-13-01409-f003:**
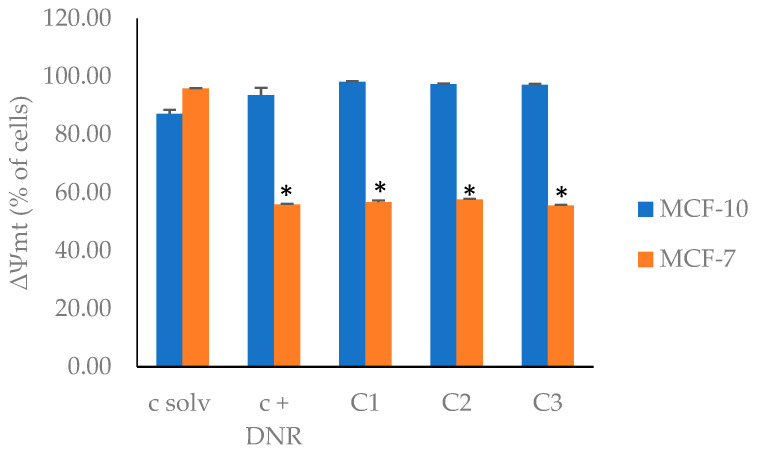
Mitochondrial membrane permeability (∆Ψmt) changes in MCF-7 and MCF-10A cell lines induced by EF2 extract. c solv: solvent control (EtOH); c + DNR: daunorubicin; C1: 50 µg/mL; C2: 100 µg/mL; C3: 200 µg/mL. * corresponds to significant differences among treatments and c solv.

**Figure 4 plants-13-01409-f004:**
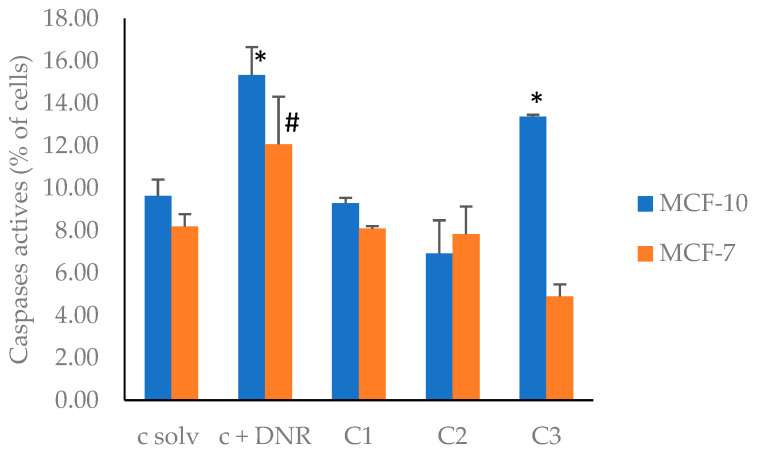
Caspase activation in MCF-7 and MCF-10A cell lines induced by EF2 extract. c solv: solvent control (EtOH); c + DNR: daunorubicin; C1: 50 µg/mL; C2: 100 µg/mL; C3: 200 µg/mL. *, #: different symbols correspond to significant differences among treatments and c solv.

**Table 1 plants-13-01409-t001:** Phytochemical contents of ethanolic extracts EF1 and EF2 of *P. chilensis*.

Ethanolic Extracts	Total Phenols (mg GAE/g DW)	Total Flavonoids (mg QE/g DW)
EF1	63.17 ^a^ ± 3.55	12.12 ^a^ ± 0.48
EF2	39.77 ^b^ ± 0.24	19.43 ^b^ ± 0.25

Mann–Whitney test was used to analyze the data. ^a,b^ Different letters correspond to significant differences among the extracts (*p* < 0.05). All data are expressed as mean ± SD (n = 3). Total phenol content is expressed as mg of gallic acid equivalents (GAEs) per gram of dry plant material weight. Total flavonoid content is expressed as mg of quercetin equivalents (QEs) per gram of dry weight.

**Table 2 plants-13-01409-t002:** Antioxidant activity of ethanolic extracts EF1 and EF2 of *P. chilensis* and positive controls.

Ethanolic Extracts/Samples	DPPH (IC_50_ mg L^−1^)	ABTS (TEAC mM)	FRAP (TEAC mM)
EF1	1.90 ± 0.12 ^a^	0.11 ± 0.01 ^a^	0.0012 ± 0.0018 ^a^
EF2	1.71 ± 0.06 ^a^	0.16 ± 0.01 ^a^	0.0014 ± 0.0001 ^a^
Trolox	0.11 ± 6.09	n.a.	n.a.
Gallic acid	2.06 ± 0.03 ^b^	1.13 ± 0.01	1.72 ± 0.02
BHT	0.06 ± 2.31	1.06 ± 0.02	1.52 ± 0.07

The Kruskal–Wallis ANOVA test was used to analyze the results. ^a,b^ Different letters correspond to significant differences among the extracts and reference compounds by antioxidant activity assay (*p* < 0.05). All data are expressed as mean ± SD (n = 3). DPPH is expressed as IC_50_ (mg L^−1^). ABTS and FRAP are expressed as TROLOX^®^ equivalents (mM).

**Table 3 plants-13-01409-t003:** IC_50_ values (µg/mL) of ethanolic extracts EF1 and EF2 of *P. chilensis* against the non-tumoral MCF-10A cell line, and cancer cell lines MCF-7 and HT-29. SI: selectivity index.

Ethanolic Extracts	MCF-10A	MCF-7	SI	HT-29	SI
EF1	>200	>200	<1	>200	<1
EF2	>200	111.25 ± 23.57	>1.8	>200	<1

**Table 4 plants-13-01409-t004:** IC_50_ values (µg/mL) of fractions obtained from EF2 of *P. chilensis* against MCF-7 and MCF-10A cell lines. HF: hexane fraction; DF: dichloromethane fraction; AF: ethyl acetate fraction; MF: methanolic fractions; QF: aqueous fraction; SI: Selectivity index. Selectivity index (SI) was calculated as the ratio of IC_50_ MCF-10A to IC_50_ MCF-7).

Fractions	MCF-7	SI	MCF-10A
HF	35.3 ± 1.5	1.3	45.0 ± 3.5
DF	107.1 ± 3.4	>2.2	232.5 ± 36.6
AF	105 ± 4.4	1.3	133.7 ± 4.4
MF	>350	<1	>350
QF	>350	<1	>350

**Table 5 plants-13-01409-t005:** GC–MS analysis for the dichloromethane fraction (DF) of *P. chilensis (f)*.

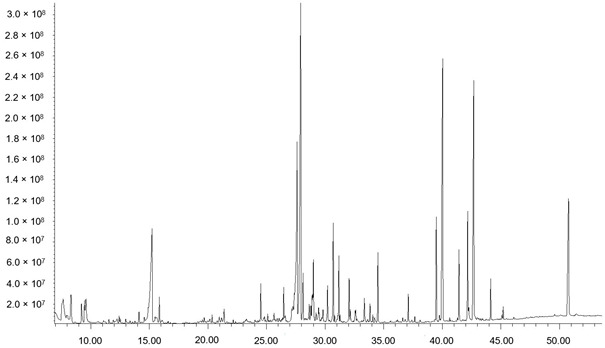							
No	RT (min)	Main Components	RI ^a^	RI ^b^	Match	Molecular Formula	% Area
1	9.467	1,3-propanediol	1053	1073	928	C_3_H_8_O_2_	0.48
2	14.125	Urea	1169	1243	899	CH_4_N_2_O	0.19
3	27.920	D-pinitol	1762	1815	894	C_7_H_14_O_6_	7.11
4	29.002	Myoinositol	1819	1930	913	C_6_H_12_O_6_	1.17
5	30.701	β-D-allopyranose	1912	1829	902	C_6_H_12_O_6_	1.32
6	31.176	Palmitic acid	1939	2039	940	C_16_H_32_O_2_	0.88
7	34.512	Stearic acid	2235	2236	926	C_18_H_36_O_2_	0.87
8	37.104	Myristic acid	2399	2424	899	C_14_H_28_O_2_	0.31
9	41.430	Sucrose	2696	2610	950	C_12_H_22_O_11_	0.89
10	42.168	2-monostearin	2751	2775	895	C_21_H_42_O_4_	1.31

^a^ RI: retention indexes relative to C_7_–C_40_
*n*-alkanes on the BP-5MS capillary column. ^b^ Retention index reported in the literature.

## Data Availability

Data are contained within the article and Appendix A.

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
