# Peer review of "Ethanolic Extract from Fruits of Pintoa chilensis, a Chilean Extremophile Plant. Assessment of Antioxidant Activity and In Vitro Cytotoxicity"

_plants, 2024, doi:10.3390/plants13101409_

Round 1

Reviewer 1 Report

Comments and Suggestions for Authors

- Lack of in vivo studies to validate the cytotoxic effects observed in vitro.

- Limited exploration of the underlying mechanisms of action of the bioactive compounds identified in the dichloromethane fraction.

- The study does not provide information on the long-term effects or potential side effects of using the P. chilensis fractions for anticancer purposes.

- The selectivity index (SI) values reported for the fractions could be further validated with additional experiments to confirm the specificity towards cancer cell lines.

-  What compounds show selective cytotoxicity?

Results section concerning the explanation of the results exhibits similar issues as previously noted. The narrative in the results section (and conclusion) is difficult to follow, and the conclusions drawn appear to diverge significantly from what can be inferred from the empirical findings.

- The discussion should prioritize organization around arguments rather than simply detailing specifics without offering substantial significance.

- Furthermore, there is a noticeable lack of crucial understanding evident in both the introduction and discussion sections. It is recommended to expand upon and cite the following study:

i. doi.org/10.1016/j.heliyon.2023.e19454

ii. 10.3390/molecules27175686

iii. doi.org/10.3389/fphar.2022.943967

iv. 10.3390/molecules27217405

v. Multidrug Resistance in Cancer: Understanding Molecular Mechanisms, Immunoprevention, and Therapeutic Approaches

vi. Berberine as a potential anticancer agent: A comprehensive review

vii. Natural small molecules in breast cancer treatment: understandings from a therapeutic viewpoint

Comments on the Quality of English Language

Major revisions

Author Response

  • Lack of in vivo studies to validate the cytotoxic effects observed in vitro.
  • R.- We agree with this reviewer in the fact that there are not in vivo studies, but these studies are far away from the main objectives of this work. Probably, its concern was induced by the title of our work that mentioned “Anticancer activity…”. However, the main goals of this work were to explore potential biological activities of the extract, including cytotoxicity effect on some cancer cell lines. An anticancer study involves in vivo studies that could be developed in the future, but it is not part of this article. Thus, to avoid confusions we have changed the title of our work, which now is “Ethanolic extract from fruits of Pintoa chilensis, a Chilean extremophile plant. Assessment of antioxidant activity and in vitro cytotoxicity”.
  • Limited exploration of the underlying mechanisms of action of the bioactive compounds identified in the dichloromethane fraction.
  • R.- This point is important, and the discussion has been improved adding some specific paragraphs in lines 397-443.
  • Lines 397-400. Here we describe action mechanism of palmitic acid, one of the compounds identified on the tested extract.
  • Lines 400-403. A similar description and discussion of available data for stearic acid has been added.
  • Lines 404-415. Mechanism of β-D-allopyranose and polyols, as D-pinitol are discussed in this paragraph.
  •  - The study does not provide information on the long-term effects or potential side effects of using the P. chilensis fractions for anticancer purposes.
  • R.- As established above, the evaluation of biological activity of the Pintoa chilensis extract have been performed through in vitro bioassays. Thus, the study of long-term and/or side effects is beyond the goal of this work.
  • The selectivity index (SI) values reported for the fractions could be further validated with additional experiments to confirm the specificity towards cancer cell lines.
  • R.- The method used for evaluation of selectivity indexes is widely used, and it is associated to the mean effective concentration. Moreover, our research project is aimed to isolation of secondary metabolites and identification of the most bioactive fraction to follow with purification and isolation of active agents by chromatographic techniques. So, at this stage of our research we think that this method is enough to select the most active fractions.
  •  What compounds show selective cytotoxicity?
  • R.- The results presented in this work do not correspond to isolated secondary metabolites. Results presented in this work correspond to cytotoxic activity of the ethanolic extract and dichloromethane fraction (DF). However, we have included in the discussion a detailed description of the main metabolites presents in DF, with the aim to correlate the observed cytotoxic activity with the extract composition.
  •  Results section concerning the explanation of the results exhibits similar issues as previously noted. The narrative in the results section (and conclusion) is difficult to follow, and the conclusions drawn appear to diverge significantly from what can be inferred from the empirical findings.
  • R.- We have rewritten the discussion adding new references and information about possible mechanisms of action. We think that the modified discussion is much more clear. Lines 351-357 and 361-370.
  • The discussion should prioritize organization around arguments rather than simply detailing specifics without offering substantial significance. Furthermore, there is a noticeable lack of crucial understanding evident in both the introduction and discussion sections. It is recommended to expand upon and cite the following study:
  • R.- We have included most of the suggested references, and we thank this reviewer for calling our attention to these studies.

Reviewer 2 Report

Comments and Suggestions for Authors

After reading the article Anticancer potential and antioxidant activity of ethanolic extract from fruits of Pintoa chilensis, a Chilean extremephile plant, I found the results interesting, however, I had some doubts.

1. The authors should attach information about Pintoa chilensis related to the objective of the research in the introduction. What information is there about Pintoa chilensis in traditional medicine?

2. Why were the concentrations 50, 100 and 200 mg/mL chosen and not others?

3. Please attach the statistics that were used to the figures and tables.

4. Please discuss at the molecular level the results with the active ingredients that were found with the dichloromethane fraction (DF) of P. chilensis (f).

5. Attach the doi to the references that contain it.

6. The authors mention that there may be synergism due to the active ingredients, in this sense, explain how this effect can originate?

Comments on the Quality of English Language

Minor editing of English language required

Author Response

The authors should attach information about Pintoa chilensis related to the objective of the research in the introduction. What information is there about Pintoa chilensis in traditional medicine?

R.- A paragraph has been added to the introduction, Lines 95-101, where the limited background information available is described. Currently, there is not information on medicinal use of this plant.

Why were the concentrations 50, 100 and 200 mg/mL chosen and not others? 

R.- These concentrations were chosen associated to the value of EF2 IC50 (111.25 ± 23.57 µg/mL), we use the IC50, a mean value and a double value for analyze the level of ROS, Mitochondrial Membrane Permeability, and active caspases. Furthermore, we consider what was described by Manosroi et al. (2006) who suggest that extracts with IC50 values less than 125 µg/mL could be possible candidates for the development of therapeutic agents against cancer.

 Please attach the statistics that were used to the figures and tables. 

R.- According to the request, the statistics used are attached to each of the tables and figures.

Please discuss at the molecular level the results with the active ingredients that were found with the dichloromethane fraction (DF) of P. chilensis (f). 

R.- We have improved the discussion by considering the possible mechanisms of action through which the main metabolites present in the dichloromethane fraction of P. chilensis could exert their action.

Attach the doi to the references that contain it. 

R.- Done

The authors mention that there may be synergism due to the active ingredients, in this sense, explain how this effect can originate? 

R.- A paragraph discussing this effect has been added at the end of the discussion section. Lines 371-376.

Round 2

Reviewer 1 Report

Comments and Suggestions for Authors

Some of my comments have been addressed. In advance to some point, the introduction and discussion still lack a crucial understanding. This can be elaborated on in the introduction and discussion. It is insisted on elaborate and cite the following study:

i. doi.org/10.1016/j.heliyon.2023.e19454

ii. 10.3390/molecules27175686

iii. doi.org/10.3389/fphar.2022.943967

iv. 10.3390/molecules27217405

v. Multidrug Resistance in Cancer: Understanding Molecular Mechanisms, Immunoprevention, and Therapeutic Approaches

vi. Berberine as a potential anticancer agent: A comprehensive review

vii. Natural small molecules in breast cancer treatment: understandings from a therapeutic viewpoint

Comments on the Quality of English Language

Major revisions

Author Response

Some of my comments have been addressed. In advance to some point, the introduction and discussion still lack a crucial understanding. This can be elaborated on in the introduction and discussion. It is insisted on elaborate and cite the following study:

doi.org/10.1016/j.heliyon.2023.e19454

10.3390/molecules27175686

iii. doi.org/10.3389/fphar.2022.943967

10.3390/molecules27217405

Multidrug Resistance in Cancer: Understanding Molecular Mechanisms, Immunoprevention, and Therapeutic Approaches

Berberine as a potential anticancer agent: A comprehensive review

Natural small molecules in breast cancer treatment: understandings from a therapeutic viewpoint

R.- From this list we have included references ii, iii, iv, vi and vii. (Ref. 8, 9, 20, 21 and 108). The others two were not considered because they are a little far away from the subject of this work. Anyway, we thank this reviewer for calling our attention on these reviews, because we consider that with addition of these references both the Introduction and Discussion have been improved.

Reviewer 2 Report

Comments and Suggestions for Authors

The Authors have taken all remarks in consideration. All issues have been clarified and the paper quality has surely improved. I therefore suggest publication.

Comments on the Quality of English Language

Minor editing of English language required

Author Response

Minor editing of English language required.

R.- English language was edited in the whole text.